# Antigen-Induced IL-1RA Production Discriminates Active and Latent Tuberculosis Infection

**DOI:** 10.3390/microorganisms11061385

**Published:** 2023-05-25

**Authors:** Cesar Sanchez, Luis Jaramillo-Valverde, Silvia Capristano, Gilmer Solis, Alonso Soto, Julio Valdivia-Silva, Julio A. Poterico, Heinner Guio

**Affiliations:** 1Laboratorio de Biotecnología y Biología Molecular, Instituto Nacional de Salud, Lima 15046, Peru; cesarsanz3@gmail.com (C.S.);; 2Escuela de Posgrado, Universidad Cesar Vallejo, Lima 15314, Peru; 3School of Medicine, Universidad Continental, Lima 15046, Peru; 4Hospital Nacional Hipólito Unanue, Lima 15007, Peru; 5INBIOMEDIC Research and Technological Center, Lima 15046, Peru; 6Centro de Investigación en Bioingeniería, Universidad de Ingenieria y Tecnologia—UTEC, Lima 15063, Peru; 7Faculty of Health Sciences, Universidad de Huánuco, Huánuco 10001, Peru; 8Centro de Investigación en Biodiversidad para la Salud, Universidad Privada Norbert Wiener, Lima 15046, Peru

**Keywords:** *Mycobacterium tuberculosis*, biomarkers, active tuberculosis, latent tuberculosis, immune response, IL-1RA

## Abstract

The IGRA (Interferon Gamma Release Assays) test is currently the standard specific test for *Mycobacterium tuberculosis* infection status. However, a positive test cannot distinguish between active tuberculosis disease (ATBD) and latent tuberculosis infection (LTBI). Developing a test with this characteristic is needed. We conducted longitudinal studies to identify a combination of antigen peptides and cytokines to discriminate between ATBD and LTBI. We studied 54 patients with ATBD disease and 51 with LTBI infection. Cell culture supernatant from cells stimulated with overlapping *Mycobacterium tuberculosis* novel peptides and 40 cytokines/chemokines were analyzed using the Luminex technology. To summarize longitudinal measurements of analyte levels, we calculated the area under the curve (AUC). Our results indicate that in vitro cell stimulation with a novel combination of peptides (Rv0849-12, Rv2031c-14, Rv2031c-5, and Rv2693-06) and IL-1RA detection in culture supernatants can discriminate between LTBI and ATBD.

## 1. Background

Tuberculosis (TB) is a communicable disease caused by the bacteria *Mycobacterium tuberculosis* (MTB) that most often affects the lungs and is spread from person to person through the air [1]. According to the Global TB Report of 2020 from the World Health Organization (WHO), TB is one of the top 10 causes of death by a single infectious agent with a total of 1.4 million deaths in 2019 worldwide [2]. Additionally, up to 2019, the WHO reported 39,000 TB cases in Peru [3]. Notably, around 30% of the Peruvian population in 2016 was estimated to have latent TB [4].

As referred, a major current stumbling block challenge in TB control is the lack of an assay to discriminate between active TB disease (ATBD) and latent tuberculosis infection (LTBI) [5]. ATBD patients have common respiratory symptoms, and radiology exams show active pulmonary disease with bacilloscopic and/or culture-positive results. In contrast, most LTBI cases are asymptomatic or oligosymptomatic and are capable to control the spread of the infection. However, in LTBI, the bacteria persist in a quiescent or latent state, becoming a potential reservoir for the development of active tuberculosis. The currently available tests, the tuberculin skin tests (TST) or interferon-γ (IFN-γ) release assays (IGRAs) show positive results in both clinical situations [5,6,7], thus, are not useful to discriminate between current ATBD and LTBI.

The TST is performed by intradermal injection of purified protein derivative (PPD) on the inner surface of the forearm. The skin test reaction is examined between 48 and 72 h after the intradermal injection to detect an induration reaction of 15 mm or larger, considered positive depending on a person’s health and exposure status. The protein derivatives PPD-SI, PPD-S2, and PPD RT23 are the most employed for this test [8]. The most significant pitfall of this test is the presence of highly conserved proteins among PPD, making this test unable to differentiate whether the positive results are because of *M. tuberculosis* infection or BCG vaccination, or exposure to environmental non-TB mycobacteria.

The IGRAs test is a whole blood assay in vitro to detect IFN-γ in supernatants produced by memory T cells stimulated with mycobacterial antigens. Antigens used include the early secretory antigenic target (ESAT6) and the 10-kDa culture filtrate protein (CFP10). These antigens are absent in the TB vaccine BCG and in most environmental mycobacteria. However, these widely used tests are not perfect and are currently used to rule out infection without the intrinsic capacity to determine whether a positive test is because of ATBD or LTBI. 

Useful to mention is the fact that the TST’s sensitivity and specificity values are between 71 and 82%, and those for IGRAs are between 81 and 86%, making IGRA tests more reliable for the diagnosis of *Mycobacterium tuberculosis* infection [9]. We considered that the careful selection of antigenic components and overall format of the IGRA tests are unique and could be improved by aiming at providing a test with the ability to discriminate between ATBD and LTBI.

For the above-mentioned reasons, it would be useful to identify antigen peptides other than those currently in use for IGRA tests and cytokines/chemokines, besides IFN-γ, that could be measured in stimulated peripheral blood mononuclear cells (PBMCs) culture supernatants and can act as potential biomarkers to distinguish between ATBD from LTBI. Thus, our study aimed at the identification of innovative antigen peptides and cytokines/chemokines useful for that purpose.

The clinical tests currently in use are prone to provide false results depending on factors such as age, nutritional and immunological status, BCG vaccination, genetic background, and cross-reactivity with environmental mycobacteria [5]. Thus, we have a priori controlled for those potential confounders. 

## 2. Patients and Methods

### 2.1. Human Subjects

Biological samples were obtained as part of the “Study of the Immunological Characteristics of Latent Tuberculosis in the Peruvian Population”. Briefly, we studied 176 participants; all of them were from Lima, Peru. A summary of the characteristics of the participants is shown in Figure 1.

We evaluated the levels of cytokines/chemokines in supernatants PBMC stimulated in vitro with antigen peptides. For this, we used PBMCs from 119 patients with ATBD, 47 with LTBI, and 39 with healthy controls. We excluded patients with TB recurrence or with reinfection with a different *Mycobacterium tuberculosis* strain; HIV infection or primary immunodeficiency; body mass index (BMI) < 18 kg/m^2^ before the disease; immunosuppressive therapy; past illnesses such as other lung diseases, cancer, autoimmunity, organ system failure, organ transplant recipient, or endocrine disease; hematologic disorders; drug misuse and addiction.

The study was approved by the Ethics Committee of Peru’s National Institute of Health. Written informed consent was obtained from all participants involved in this study. We aimed to collect these samples from patients diagnosed with active TB disease before TB treatment initiation. 

### 2.2. Antigens

Peptides of 20 amino acids in length with 10 amino acids overlapping sequences were synthesized, purified by Standard Chemical Peptide Synthesis, and analyzed and purified by mass spectrometry and HPLC (GenScript). Peptides were dissolved in DMSO (10 μg/mL final concentration) and stored at −80 °C until used. 

We used peptides covering sequences of *Mycobacterium tuberculosis* antigens: Rv0849 (Rv0849-11, Rv0849-12, Rv0849-13, Rv0849-14, Rv0849-15, Rv0849- 16, Rv0849-17, Rv0849-18, Rv0849-19, Rv0849-20); Rv1986 (Rv1986-01, Rv1986-1b, Rva1986-02, Rv1986-03, Rv1986-04, Rv1986-05, Rv1986-06, Rv1986-07, Rv1986-08, Rv1986-09, Rv1986-10, Rv1986-11, Rv1986-12, Rv1986-13, Rv1986-14, Rv1986-15, Rv1986-16, Rv1986-17, Rv1986-18, Rv1986-19); Rv2693 (Rv2693c-01, Rv2693c-01b, Rv2693c-02, Rv2693c-03, Rv2693c-04, Rv2693c-05, Rv2693c-06, Rv2693c-07, Rv2693c-08, Rv2693c-09, Rv2693c-10, Rv2693c-11); Rv2031 (Rv2031c-01, Rv2031c-02, Rv2031c-03, Rv2031c-04, Rv2031c-05, Rv2031c-06, Rv2031c-07, Rv2031c-08, Rv2031c-09, Rv2031c-10, Rv2031c-11, Rv2031c-12, Rv2031c-13, Rv2031c-14); ESAT6 (early secretory antigenic target 6 kD).

### 2.3. PBMC Isolation and Peptide Stimulation

Peripheral blood mononuclear cells (PBMC) were obtained from heparinized blood samples isolated by density gradient centrifugation using Ficoll–Hypaque (GE General Electric, Chicago, IL, USA). Cell viability was determined by trypan blue exclusion. For cell culture, PBMC (2 × 10^5^/mL) was resuspended in RPMI 1640 medium with L-glutamine and sodium bicarbonate (Gibco), supplemented with 10% of heat-inactivated fetal bovine serum (Gibco, Hampton, NH, USA). PBMCs were seeded in 96-well plates with 100 μL of each peptide (final concentration 10 μg/mL) and incubated for 72 h at 37 °C in 5% CO_2_. Phytohemagglutinin (PHA M-form, concentration 1.5% *v*/*v*, Gibco), and dimethyl sulfoxide (DMSO, concentration 100%, OriGen Biomedical, Austin, TX, USA) were included as positive and negative controls, respectively. 

### 2.4. Luminex Cytokine Detection

Supernatants from cell cultures were collected after 72 h. A panel of 40 soluble immunological molecules were tested, including: EGF, FGF, EoTX, TGF, G-CSF, FRAK, IFNa2, IFNg, GRO, IL-10, MCP-3, IL-12, MDC, IL-12 p70, IL-13, IL-15, sCD40L, IL-17, IL-1RA, IL-1a, IL-9, IL1-B, IL-2, IL-3, IL-4, IL-5, IL-6, IL-7, IL-8, IP-10, MCP-1, MIP-1a, MIP-1B, TNFa, TNFB, VEGF, IL-22, MCP-2. Detection of those analytes was conducted using Luminex assays (Merck, Darmstadt, Germany). Assays were performed according to the manufacturer’s instructions. Data were collected and analyzed using Luminex software (Bio-Plex Manager Software, Standard Edition). A five-parameter regression formula was used to calculate the sample concentration from the standard curves. 

### 2.5. Statistical Analysis

#### 2.5.1. Normality Test

The normality test was performed for each variable. The Shapiro–Wilk and Kolmogorov–Smirnov test were used. All variables had a non-normal distribution. In addition, descriptive statistics for non-parametric models were used.

#### 2.5.2. Bivariate and Multivariate Analysis

To determine whether or not there is a statistically significant difference between two quantitative variables, we used the non-parametric U-Mann–Whitney test or the non-parametric Kruskal–Wallis test for more than two quantitative variables groups. When a significant *p*-value was observed in the Kruskal–Wallis test, the Dunnett post hoc test was performed to evaluate the difference between the groups at the 0.05 significance level. We performed diagnostic test analysis combining cytokines and peptides using binary logistic regression. Data analysis was performed using Stata 16.1 software (Stata Corporation, College Station, TX, USA).

#### 2.5.3. The Area under the Curve (AUC)

We used the area under the ROC curve (denoted AUC) to provide a measure of the model’s ability to discriminate between active TB disease (ATBD) and latent TB infection (LTBI). We used the c-statistic to compare all possible pairs of individuals consisting of one individual who experienced ATBD and one individual who experienced LTBI outcome. Thus, the c-statistic is the proportion of such pairs in which the individual who experienced the ATBD outcome had a higher predicted probability of experiencing the outcome than the individual who experienced the LTBI outcome. A value of 0.5 (50%) indicates no ability to discriminate while a value of 1 (100%) indicates a perfect ability to discriminate.

## 3. Results

The clinical characteristics of the study participants are summarized in Table 1. 

First, we conducted a pilot analysis using samples from 15 LTBI and 15 ATBD participants. The objective was to select the most immunogenic set of peptides for the analysis of the entire sample. We identified four peptides: Rv0849-12, Rv2031c-14, Rv2031c-5, and Rv2693-06 as the most immunogenic. PBMCs from all participants were stimulated with those peptides. Table 2 shows the data obtained from LTBI and ATBD. Data shown were obtained by subtracting levels of cytokines/chemokines observed in the nil (no antigen-stimulated variable) to the levels observed for each active variable in each group. We observed significantly higher levels of IL1-RA in supernatants of peptide-stimulated PBMCs obtained from ATBD than in supernatants obtained from LTBI. This observation was consistently true when cells were stimulated with peptides Rv0849-12, Rv2031c-14, Rv2031c-5, and Rv2693-06. Other cytokines/chemokines produced at important levels were IP-10, MCP-1, IL-8, MCP-3, GRO, EGF, MDC, and IL-13, but their levels were not consistently high or low when cells were stimulated with the four peptides mentioned above. These results were summarized in Table 3 and Figure 2, Figure 3, Figure 4 and Figure 5. 

We then calculated the area under the curve (AUC) for each analyte to determine which cytokine/chemokine and set of peptides had a better capacity to discriminate between LTBI and ATBD. We used the AUC because this is the definite integral of a curve that describes the variation of an analyte concentration as a function of time. Once again, the IL-1RA analyte showed the best performance as only IL-1RA served as a discriminative analyte when cells were stimulated with each of the four peptides and in the highest proportion (76.81%) of cases (Figure 6).

## 4. Discussion

We conducted this study to obtain a combination of peptides and cytokines/chemokines that can help us to discriminate between the LBTI and ATBD. For this purpose, we carefully chose the participants based on clinical, demographic, and epidemiological information and appropriate laboratory/radiological tests. This selection process was laborious but necessary to deliver conclusions with meaningful clinical significance. 

The peptides we used were previously selected among antigens that have shown to be immunogenic under hypoxic conditions, a condition observed in the granulomatous and caseating granuloma foci. Regarding antigen selection, our main goal was to obtain a combination of those antigen peptides that can serve to stimulate cytokines/chemokines and their preferential expression in ATBD or LTBI cases. We used overlapping peptides of 20 residues long to ensure the finding of the best epitopes [5,10,11].

In our study, IL-1RA emerged as the most useful analyte to discriminate between ATBD and LTBI, as we found it at significantly higher levels in supernatants of cells from ATBD than those from LTBI when the cells were stimulated with any of the four peptides selected. Others, such as IP-10, MCP-1, IL-8, MCP-3, GRO, EGF, MDC, and IL-13 had mixed inconsistent results for each of the peptides analyzed. 

IL-1RA is an IL-1 antagonist produced during the inflammatory process. IL1-RA has been found to be associated with increasing susceptibility to TB infection and was found to be increased in active TB [12]. Interestingly, IL-1RA was detectable in the serum of LTBI cases and detected at lower levels in LTBI-treated cases than in untreated cases, arguing in favor of its use as a marker for active disease development [13,14]. Although this protein inhibits the activities of interleukin 1, alpha (IL1α) and interleukin 1, and beta (IL1β), in TB, it seems to have a modulatory role in the inflammatory response mediated by these cytokines. Several studies that demonstrate the importance of IL1 in the immune response against intracellular bacteria show that an exacerbated increase in its levels ends in significant tissue damage [15,16]. The presence of inhibitors such as IL1RA, therefore, suggests an important step in the process demonstrated by its high expression in active disease and during antigenic stimulation of white cells isolated from patients as shown in this study. Noteworthy, the role of IL-1RA in TB pathophysiology requires further study [17].

Our findings are relevant because approximately 2 billion people live with latent TB. In other words, about 25% of the world’s population is latently infected with *Mycobacterium tuberculosis* and at risk of developing ATBD as 5–10% of them will develop active disease in their lifetime [2]. Thus, the detection and appropriate treatment of LTBI cases will have a huge impact on TB elimination.

The peptides Rv0849, Rv2031c, Rv2031, and Rv2693 are involved in various aspects of MTB pathogenesis and have immunogenic properties, which could make them useful in stimulating IL-1RA production. In the context of ATBD and LTBI, it is possible that these peptides differentially stimulate IL-1RA production, thus contributing to the distinction between these two infection statuses. Some studies have also shown that Rv0849 induces the secretion of pro-inflammatory cytokines, such as IFN-γ, and may be involved in the granuloma formation [11,18,19]. Rv2031c plays a role in maintaining the persistence of MTB in the host and has been associated with immune evasion strategies, including the suppression of antigen presentation and the modulation of host cytokine responses, such as downregulating TNF-α production [20]. The Rv2031 protein is reported to play a role in modulating the host immune response, specifically by inhibiting the production of reactive oxygen species (ROS) and promoting the survival of MTB in macrophages [21,22]. Moreover, Rv2693, in its involvement in IL-1RA stimulation suggests that it may play a role in modulating host immune responses during MTB infection [23].

The balance between pro- and anti-inflammatory cytokines, including IL-1RA, is crucial for determining the outcome of MTB infection [13,24]. Research has shown that IL-1RA levels may be differentially regulated in ATBD and LTBI. For instance, a study found that IL-1RA levels were significantly higher in the culture supernatants of peripheral blood mononuclear cells (PBMCs) from patients with ATBD compared to those with LTBI. This suggests that IL-1RA may play a role in discriminating between ATBD and LTBI [25]. Other research is required to elucidate the precise mechanisms by which these peptides interact with the host immune system to stimulate IL-1RA production and how this relates to the pathogenesis of active and latent TB.

Our study has some limitations. We included 119 patients with ATBD and 47 with LTBI, and while these numbers provided preliminary insights into the potential of our method to discriminate between ATBD and LTBI, a larger and more diverse patient cohort would be necessary to confirm and generalize our findings [26]. Although our study employed a longitudinal design to capture dynamic immune responses, the data may not fully reflect the potential variability in individual responses over time or account for the impact of various factors such as immune status, previous infectious diseases, etc. [27]. Moreover, our method relies on Luminex technology to analyze cytokines/chemokines, which may not be feasible in resource-limited settings where TB is highly prevalent [28]. Alternative, low-cost, and easy-to-use diagnostic tools would be desirable for broader implementation in such contexts [29].

For this aforementioned reason, the new method’s performance needs to be validated in different populations to account for potential differences in immune responses and MTB strains especially in Perú that has a large population living in high altitude and in the Amazon jungles [30]. Prospective studies are necessary to further validate our method’s accuracy and clinical relevance in discriminating between ATBD and LTBI, as well as to monitor the effectiveness of TB treatment [31].

## 5. Conclusions

The detection of IL-1RA in supernatants of PBMCs stimulated with this novel combination of *Mycobacterium tuberculosis* antigen peptides (Rv0849-12, Rv2031c-14, Rv2031c-5, and Rv2693-06) can discriminate ATBD and LBTI. This is an important finding, especially in settings where TB prevalence is high.

## Figures and Tables

**Figure 1 microorganisms-11-01385-f001:**
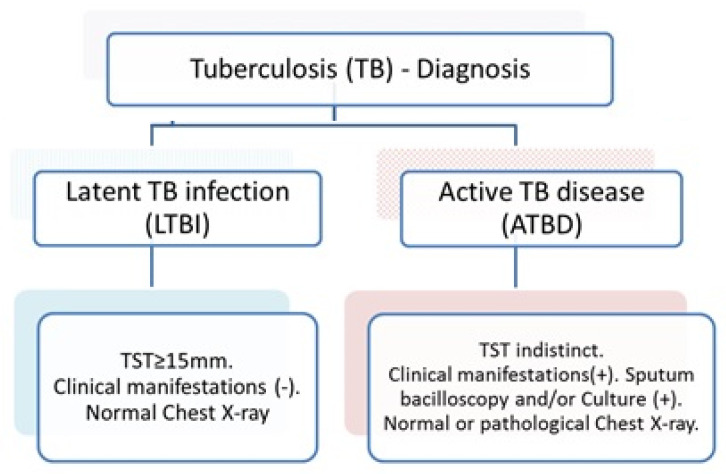
Eligibility criteria for the different participant groups included in the study. Number of samples = 30.

**Figure 2 microorganisms-11-01385-f002:**
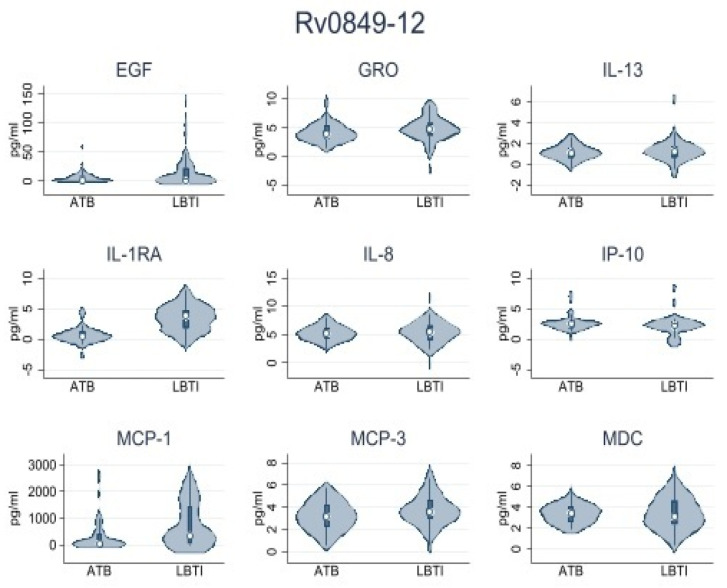
Rv0849-12 peptide. Cytokine/chemokine levels analyzed by Luminex. Number of samples = 30.

**Figure 3 microorganisms-11-01385-f003:**
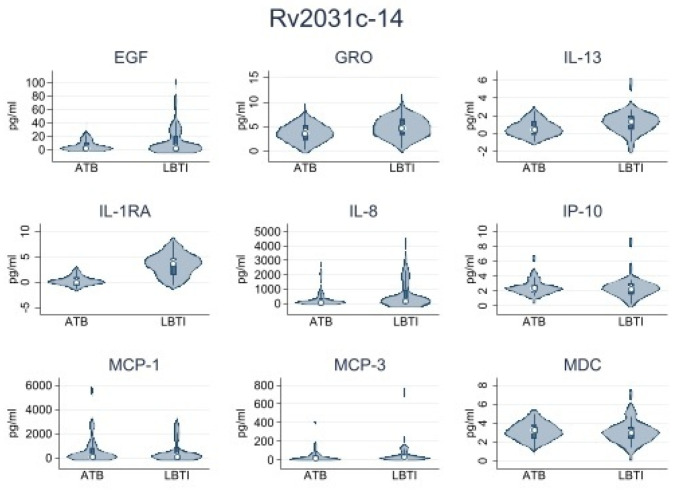
Rv2031-14 peptide. Cytokine/chemokine levels analyzed by Luminex. Number of samples = 30.

**Figure 4 microorganisms-11-01385-f004:**
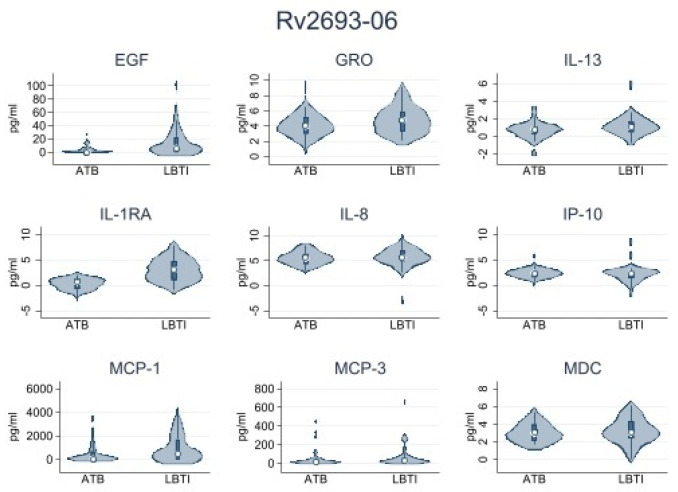
Rv2693-06 peptide. Cytokine/chemokine levels analyzed by Luminex. Number of samples = 30.

**Figure 5 microorganisms-11-01385-f005:**
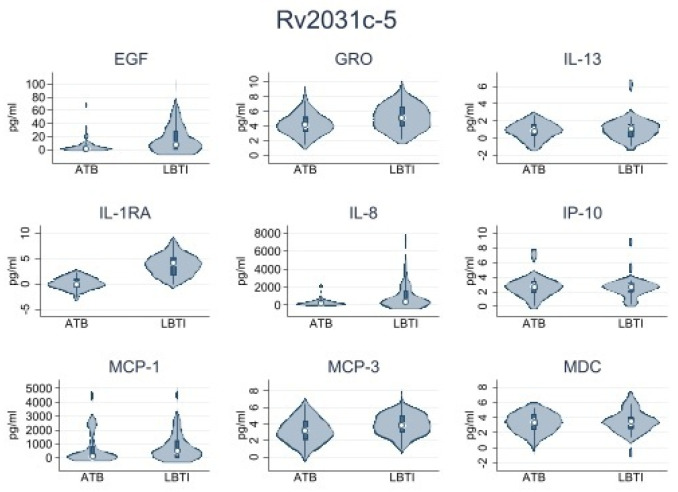
Rv2031c-5 peptide. Cytokine/chemokine levels analyzed by Luminex. Number of samples = 30.

**Figure 6 microorganisms-11-01385-f006:**
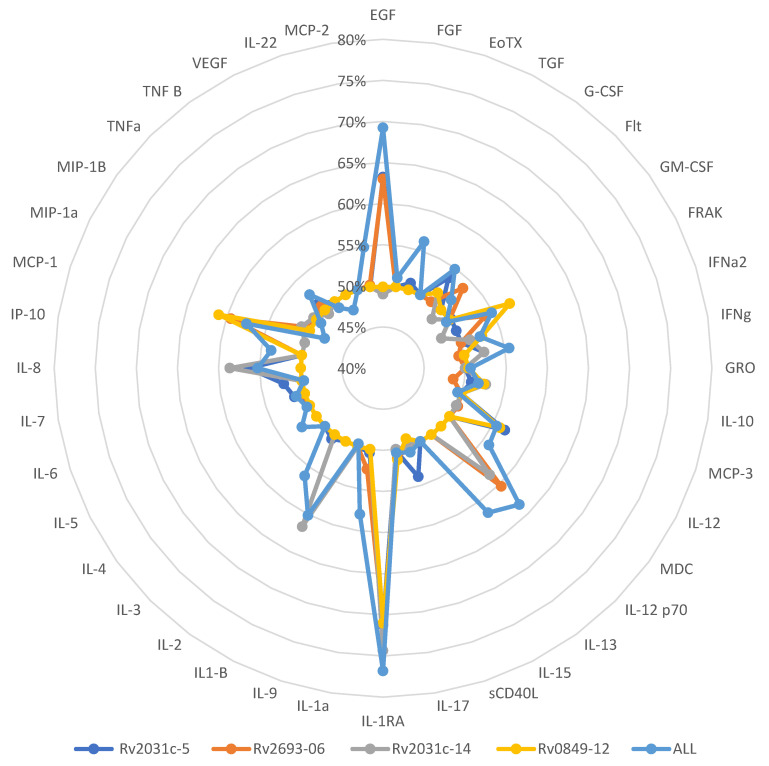
Combination of Cytokines and Peptides and their performance results (c-statistic). Number of samples = 166.

**Table 1 microorganisms-11-01385-t001:** Demographic baseline characteristics of study participants.

Characteristics	ATBD (%)	LTBI (%)	Total n	*p*-Value
Gender				
Male	73 (77.7)	21 (22.3)	94	0.051 ^†^
Female	46 (63.9)	26 (36.1)	72	
Age				
12–17 years	1 (50)	1 (50)	2	<0.001 ^‡^
18–29 years	76 (87.4)	11 (12.6)	87	
30 or more years	42 (54.5)	35 (45.5)	77	
Ethnicity				
Caucasian	6 (75.0)	2 (25.0)	8	1.000 ^‡^
Mestizo	113 (71.5)	45 (28.5)	158	
Marital status				
Single	63 (82.9)	13 (17.1)	76	0.008 ^‡^
Married	17 (54.8)	14 (45.2)	31	
Cohabiting	0 (0.0)	1 (100)	1	
Separated	2 (66.7)	1 (33.3)	3	
Divorced	37 (67.3)	18 (32.7)	55	
Education				
None	1 (100)	0 (0.0)	1	<0.001 ^‡^
Primary	7 (70.0)	3 (30.0)	10	
High school	99 (83.9)	19 (16.1)	118	
Technical	9 (34.6)	17 (65.4)	26	
University degree	3 (27.3)	8 (72.7)	11	
Work				
Unemployed	2 (100)	0 (0.0)	2	<0.001 ^‡^
Student	13 (72.2)	5 (27.8)	18	
House worker	29 (90.6)	3 (9.4)	32	
Temporal worker	7 (70.0)	3 (30.0)	10	
Health care worker	2 (11.1)	16 (88.9)	18	
Public/private worker	42 (80.8)	10 (19.2)	52	
Independent worker	24 (70.6)	10 (29.4)	34	
Comorbidities				
None	80 (71.4)	32 (28.6)	112	1.000 ^‡^
Yes	39 (72.2)	15 (27.8)	54	
Total	119 (71.7)	47 (28.3)	166	

Number of samples = 166. ^†^ Pearson’s Chi Square Test. ^‡^ Fisher’s Exact Test.Teenag.

**Table 2 microorganisms-11-01385-t002:** Significant levels of Cytokines and Peptides combined in pilot study.

Cytokines and Peptides Combined	LTBI	ATBD	*p*-Value *
Rv0849-12			
IL-1RA	0.48 (0.0–1.87)	5.57 (0.0–84.45)	0.001
IP-10	8.72 (0.0–15.01)	1.09 (0.0–10.60)	0.054
MCP-1	46.87 (0.0–408.52)	361.73 (48.60–1447.92)	0.003
Rv2031c-14			
IL-1RA	0.51 (0.0–1.18)	13.55 (0.26–73.2)	0.001
Rv2031c-5			
IL-1RA	0.47 (0.0–1.42)	4.51 (0.0–84.50)	0.006
IL-8	189.6 (26.91–441.22)	368.65 (92.22–1611.75)	0.041
MCP-3	14.13 (2.23–45.15)	30.53 (9.46–132.49)	0.033
GRO	45.57 (8.05–139.76)	105.35 (21.05–579.55)	0.017
EGF	1.57 (0.0–6.71)	7.78 (0.0–28.99)	0.030
Rv2693-06			
IL-1RA	0.06 (0.0–1.87)	4.78 (0.0–69.27)	0.001
IL-8	168.23 (43.61–368.22)	266.01 (123.42–1031.04)	0.038
MCP-1	42.15 (0.0–656.25)	468.47 (0.0–1689.57)	0.029
GRO	33.11 (9.28–94.95)	77.50 (21.21–259.76)	0.036
EGF	0 (0.0–4.95)	6.17 (0.0–22.15)	0.001
MDC	0 (0.0–7.26)	6.80 (0.0–27.95)	0.002
IL-13	0 (0.0–2.08)	1.68 (0.0–3.94)	0.022

Median (Q1–Q3), * Mann–Whitney test, Number of samples = 30.

**Table 3 microorganisms-11-01385-t003:** Cytokines and Peptides combined in main study.

	Rv0849-13			Rv2031c-6			Rv2031c-15			Rv2693-07		
	LTBI	ATBD	*p **	LTBI	ATBD	*p **	LTBI	ATBD	*p **	LTBI	ATBD	*p **
EGF	1.65 (0.0–9.04)	0 (0.0–22.73)	0.504	1.57 (0.0–6.71)	7.78 (0.0–28.99)	0.030	1.76 (0.0–10.53)	2.47 (0.0–20.19)	0.554	0 (0.0–4.95)	6.17 (0.0–22.15)	0.001
EOTAXIN	0 (0.0–3.29)	0 (0.0–2.53)	0.513	0.49 (0.0–3.31)	0 (0.0–3.29)	0.362	0.74 (0.0–2.6)	0 (0.0–3.41)	0.630	0.49 (0.0–1.7)	0 (0.0–3.29)	0.821
FGF-2	0.0 (0.0–0.0)	0 (0.0–1.96)	0.409	0.0 (0.0–0.0)	0 (0.0–1.75)	0.682	0 (0.0–1.16)	0 (0.0–2.37)	0.650	0.0 (0.0–0.0)	0.0 (0.0–0.0)	0.984
Flt	0 (0.0–1.21)	0 (0.0–1.22)	0.546	0 (0.0–0.72)	0 (0.0–1.56)	0.307	0 (0.0–1.21)	0 (0.0–1.12)	0.593	0 (0.0–1.02)	0 (0.0–0.73)	0.331
Fractalkine	11.77 (0.0–21.35)	0 (0.0–13.07)	0.113	10.96 (0.0–19.53)	8.04 (0.0–18.30)	0.549	10.39 (0.0–13.07)	0 (0.0–13.07)	0.248	11.77 (0.0–19.53)	0 (0.0–14.27)	0.081
G-CSF	0 (0.0–2.14)	0.41 (0.0–0.86)	0.117	0.41 (0.0–2.37)	0 (0.0–2.14)	0.170	0 (0.0–2.14)	0 (0.0–1.88)	0.673	0 (0.0–1.89)	0.0 (0.0–0.0)	0.072
GM-CSF	0.2 (0.0–0.92)	0.12 (0.0–1.17)	0.761	0.1 (0.0–1.22)	0.34 (0.0–1.53)	0.602	0.04 (0.0–0.89)	0.19 (0.0–1.7)	0.422	0.2 (0.0–1.33)	0.37 (0.0–1.19)	0.894
GRO	32.21 (9.47–153.24)	60.57 (1.08–230.83)	0.563	45.57 (8.05–139.76)	105.35 (21.05–579.55)	0.017	21.05 (1.52–78.48)	37.51 (3.05–466.44)	0.136	33.11 (9.28–94.95)	77.50 (21.21–259.76)	0.036
IFNa2	0.54 (0.0–2.05)	0 (0.0–2.11)	0.716	0 (0.0–1.23)	0.53 (0.0–2.27)	0.307	0 (0.0–1.23)	0.27 (0.0–1.94)	0.599	0 (0.0–1.23)	0 (0.0–1.20)	0.707
IFNg	0.0 (0.0–0.0)	0 (0.0–0.23)	0.262	0.0 (0.0–0.0)	0.0 (0.0–0.0)	0.763	0.0 (0.0–0.0)	0.0 (0.0–0.0)	0.528	0 (0.0–0.37)	0.0 (0.0–0.0)	0.166
IL-10	0 (0.0–12.43)	2.55 (0.0–16.13)	0.384	1.7 (0.0–17.37)	2.84 (0.0–13.55)	0.929	1.26 (0.0–13.42)	0.31 (0.0–12.79)	0.833	2.71 (0.0–18.44)	2.78 (6.08–61.70)	0.932
IL-12p40	0 (0.0–0.84)	0 (0.0–0.09)	0.043	0 (0.0–1.47)	0 (0.0–0.70)	0.123	0 (0.0–0.76)	0 (0.0–0.78)	0.786	0 (0.0–0.63)	0.0 (0.0–0.0)	0.425
IL-12p70	0 (0.0–0.37)	0 (0.0–0.60)	0.173	0 (0.0–0.48)	0.21 (0.0–0.96)	0.050	0 (0.0–0.49)	0.53 (0.0–1.47)	0.029	0 (0.0–0.33)	0.27 (0.0–1.28)	0.023
IL-13	0.63 (0.0–3.11)	1.42 (0.0–4.22)	0.428	1.04 (0.0–3.14)	0.39 (0.0–3.25)	0.348	0 (0.0–1.33)	0.60 (0.0–4.23)	0.117	0 (0.0–2.08)	1.68 (0.0–3.94)	0.022
IL-15	0.0 (0.0–0.0)	0.0 (0.0–0.0)	1.000	0.0 (0.0–0.0)	0.0 (0.0–0.0)	1.000	0.0 (0.0–0.0)	0.0 (0.0–0.0)	1.000	0.0 (0.0–0.0)	0.0 (0.0–0.0)	0.370
IL-17	0.0 (0.0–0.0)	0.0 (0.0–0.0)	0.849	0.0 (0.0–0.0)	0.0 (0.0–0.0)	0.710	0.0 (0.0–0.0)	0.0 (0.0–0.0)	0.293	0.0 (0.0–0.0)	0.0 (0.0–0.0)	0.442
IL-1a	0.0 (0.0–0.0)	0.0 (0.0–0.0)	0.071	0.0 (0.0–0.0)	0.0 (0.0–0.0)	0.029	0.0 (0.0–0.0)	0.0 (0.0–0.0)	0.029	0.0 (0.0–0.0)	0.0 (0.0–0.0)	0.006
IL-1B	0.0 (0.0–0.0)	0 (0.0–0.59)	0.408	0.0 (0.0–0.0)	0 (0.0–1.34)	0.121	0.0 (0.0–0.0)	0 (0.0–0.65)	0.015	0.0 (0.0–0.0)	0 (0.0–0.32)	0.025
IL-1RA	0.48 (0.0–1.87)	5.57 (0.0–84.45)	0.001	0.47 (0.0–1.42)	4.51 (0.0–84.50)	0.006	0.51 (0.0–1.18)	13.55 (0.26–73.2)	0.001	0.06 (0.0–1.87)	4.78 (0.0–69.27)	0.001
IL-2	0.0 (0.0–0.0)	0.0 (0.0–0.0)	0.426	0.0 (0.0–0.0)	0.0 (0.0–0.0)	0.649	0.0 (0.0–0.0)	0.0 (0.0–0.0)	0.373	0.0 (0.0–0.0)	0.0 (0.0–0.0)	0.937
IL-22	0.0 (0.0–0.0)	0.0 (0.0–0.0)	1.000	0.0 (0.0–0.0)	0.0 (0.0–0.0)	1.000	0.0 (0.0–0.0)	0.0 (0.0–0.0)	1.000	0.0 (0.0–0.0)	0.0 (0.0–0.0)	1.000
IL-3	0.0 (0.0–0.0)	0.0 (0.0–0.0)	0.265	0.0 (0.0–0.0)	0.0 (0.0–0.0)	1.000	0.0 (0.0–0.0)	0.0 (0.0–0.0)	1.000	0.0 (0.0–0.0)	0.0 (0.0–0.0)	1.000
IL-4	0 (0.0–6.05)	3.58 (0.0–8.24)	0.349	0 (0.0–6.05)	2.47 (0.0–13.55)	0.129	0 (0.0–8.15)	0.28 (0.0–8.73)	0.469	0.28 (0.0–6.05)	2.61 (0.0–8.45)	0.392
IL-5	0 (0.0–0.05)	0 (0.0–0.10)	0.479	0 (0.0–0.05)	0 (0.0–0.08)	0.435	0 (0.0–0.05)	0 (0.0–0.05)	0.194	0 (0.0–0.05)	0 (0.0–0.05)	0.621
IL-6	0 (0.0–1.1)	0 (0.0–0.72)	0.917	0 (0.0–3.46)	0.14 (0.0–4.77)	0.540	0 (0.0–0.57)	0 (0.0–4.06)	0.127	0 (0.0–1.11)	0 (0.0–2.02)	0.950
IL-7	1.52 (0.0–3.39)	0.72 (0.0–3.85)	0.369	1.66 (0.0–2.89)	1.46 (0.0–4.32)	0.952	1 (0.0–2.55)	0.95 (0.0–3.23)	0.838	1.03 (0.0–1.96)	1.52 (0.0–3.16)	0.930
IL-8	114.04 (30.14–404.76)	182.12 (16.62–600.10)	0.396	189.6 (26.91–441.22)	368.65 (92.22–1611.75)	0.041	66.98 (0.0–371.87)	141.5 (30.32–954.13)	0.059	168.23 (43.61–368.22)	266.01 (123.42–1031.04)	0.038
IL-9	0 (0.0–1.32)	0.92 (0.0–2.41)	0.158	0.1 (0.0–1.15)	0.63 (0.0–2.51)	0.269	0 (0.0–1.02)	0.26 (0.0–1.75)	0.267	0.59 (0.0–1.93)	0.23 (0.0–2.14)	0.691
IP-10	8.72 (0.0–15.01)	1.09 (0.0–10.60)	0.054	7.65 (1.15–20.67)	6.39 (0.0–17.13)	0.193	7.97 (0.0–12.31)	3.44 (0.0–10.50)	0.114	5.8 (0.0–16.68)	3.58 (0.0–13.96)	0.300
MCP-1	46.87 (0.0–408.52)	361.73 (48.60–1447.92)	0.003	77.6 (0.0–815.13)	465.28 (11.51–1267.78)	0.105	82.22 (0.0–819.08)	120.37 (0.0–882.54)	0.989	42.15 (0.0–656.25)	468.47 (0.0–1689.57)	0.029
MCP-2	0.64 (0.0–2.31)	1.23 (0.0–3.02)	0.517	0.94 (0.0–3.23)	1.22 (0.0–4.38)	0.961	0.64 (0.0–1.64)	1.08 (0.0–3.72)	0.540	0.53 (0.0–1.91)	0.38 (0.0–1.79)	0.683
MCP-3	14.93 (3.59–48)	27.65 (6.29–79.52)	0.268	14.13 (2.23–45.15)	30.53 (9.46–132.49)	0.033	10.89 (0.0–43.31)	26.33 (8.84–52.16)	0.078	12.92 (0.0–40.57)	28.2 (6.08–61.70)	0.079
MDC	9.66 (0.0–38.23)	9.35 (0.0–54.15)	0.474	5.48 (0.0–36.56)	20.46 (2.31–48.27)	0.091	2.68 (0.0–25.66)	7.05 (0.0–25.16)	0.512	0 (0.0–7.26)	6.80 (0.0–27.95)	0.002
MIP-1a	0.91 (0.0–11.17)	1 (0.0–12.01)	0.843	4.56 (0.0–12.92)	11.42 (0.0–27.52)	0.663	0 (0.0–3.85)	1.03 (0.0–11.36)	0.105	0 (0.0–3.09)	0 (0.0–6.49)	0.302
MIP-1B	0.71 (0.0–8.36)	|	0.965	7.09 (0.0–13.81)	8.99 (0.0–18.61)	0.547	0 (0.0–4.74)	0 (0.0–7.98)	0.502	0 (0.0–3.64)	0 (0.0–7.52)	0.621
sCD40L	3.305 (0.0–6.54)	3.71 (0.0–13.14)	0.386	2.48 (0.0–4.19)	2.94 (0.0–13.81)	0.242	1.45 (0.0–4.63)	2.52 (0.0–9.94)	0.355	0.78 (0.0–3.49)	2.88 (0.0–10.37)	0.141
TGF a	0.0 (0.0–0.0)	0.0 (0.0–0.0)	0.113	0.0 (0.0–0.0)	0.0 (0.0–0.0)	0.113	0.0 (0.0–0.0)	0.0 (0.0–0.0)	0.113	0.0 (0.0–0.0)	0.0 (0.0–0.0)	0.113
TNF B	0.06 (0.0–0.12)	0.06 (0.0–0.12)	0.696	0 (0.0–0.06)	0.06 (0.0–0.13)	0.064	0 (0.0–0.06)	0.06 (0.0–0.13)	0.029	0 (0.0–0.06)	0 (0.0–0.12)	0.361
TNFa	0.6 (0.0–4.28)	0.33 (0.0–4.01)	0.986	1 (0.0–3.61)	1.04 (0.0–3.50)	0.681	0 (0.0–1.65)	0.06 (0.0–1.87)	0.453	0.84 (0.0–4.42)	0.04 (0.0–2.23)	0.290
VEGF	1.48 (0.0–4.26)	0.33 (0.0–6.45)	0.795	0 (0.0–4.78)	0.88 (0.0–7.52)	0.345	1.14 (0.0–4.86)	0 (0.0–5.75)	0.764	1.33 (0.0–4.49)	1.02 (0.0–6.07)	0.857

Median (Q1–Q3). * Mann–Whitney test. Number of samples = 166.

## Data Availability

The data presented in this study are available on request from the corresponding author.

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
