# Peer review of "Antigen-Induced IL-1RA Production Discriminates Active and Latent Tuberculosis Infection"

_microorganisms, 2023, doi:10.3390/microorganisms11061385_

Round 1
Reviewer 1 Report
The manuscript is in a similar vein to other recent papers identifying immune correlates that are predictive of disease outcome for tuberculosis.
Overall while it has reasonable readability, there are many areas, some listed below, that need to be addressed together with proof reading by someone with good English skills, as there are numerous spelling and grammatical errors.
1> Pg 3, line 90 –We evaluated the levels of cytokines/chemokines in supernatants PBMC stimulated in vitro with antigen peptides We used PBMCs from 54 patients with ATBD? This is two different sentences, please check the reading.
2> Pg 3, line 93 body mass index (BMI) <18 kg/m2 before the disease. Can you explain why excluded patients BMI<18 kg/m?
3> Pg 5, the data in table 1 are conflict with study design in Pg 3, line 90, the study used PBMCs from 54 patients with ATBD, 51 with LTBI, 32 undetermined and 39 healthy controls. But in table 1, there were 119 active TB and 47 Latent TB.
4> Pg 6, control missing in the figure 2 to figure 5. Samples from 15 healthy controls, 15 LTBI and 15 ATBD participants were test, why there were only data for LTBI and ATBD participants, the healthy controls should be compared together.
5> Pg 8- Pg 9, The table is not complete and difficult to understand. Please adjust the layout of the table and further analysis of tabular data.
6> The article can be improved by further discussing the limitation of the study, and the constraints for the implementation of the new method in current clinical practice.
Reviewer 2 Report
The general writing style needs improvement. For instance, Mycobacterium tuberculosis needs to be in italics. Check the consistency of tuberculosis throughout the manuscript.
Check abbreviations LTBI in the abstract and in the figure legends.
The figure legends need expanding, such as the number of samples and statistical test used.
Can the authors provide a reference work in addition to the reason for their choice of antigens/peptides tested?
Can the authors include a discussion on the peptides Rv0849, Rv2031c, Rv2031 and Rv2693 they found to be relevant for IL-1RA stimulation in active TB vs latent TB. What are the physiological role of these genes/peptides in Mtb and how they may interact with the human host?
